# Health-related quality of life in Brazilian patients with cutaneous leishmaniasis using EQ-5D

Janaína de Pina Carvalho[1☉], Sarah Nascimento Silva[1☉*], Tália Santana Machado de Assis[1,2], Endi Lanza Galvão[1,3], Mayra Soares Moreira[1], Mônica Viegas Andrade[4], Kenya Valéria Micaela de Souza Noronha[4], Gláucia Cota[1]

1 Pesquisa Clínica e Políticas Públicas em Doenças Infecciosas e Parasitárias, Núcleo de Avaliação de Tecnologias em Saúde, Instituto René Rachou, Fundação Oswaldo Cruz, Belo Horizonte, Minas Gerais, Brazil, 2 Centro Federal de Educação Tecnológica de Minas Gerais, Campus Contagem, Contagem, Minas Gerais, Brazil, 3 Programa de Pós-Graduação em Reabilitação e Desempenho Funcional, Departamento de Fisioterapia, Universidade Federal dos Vales do Jequitinhonha e Mucuri, Diamantina, Minas Gerais, Brazil, 4 Centro de desenvolvimento e Planejamento Regional, Faculdade de Ciências Econômicas da Universidade Federal de Minas Gerais, Minas Gerais, Brazil

☉ These authors contributed equally to this work.
* sarah.nascimento@fiocruz.br

## Abstract

Cutaneous leishmaniasis (CL) is a neglected infectious disease with a global distribution and a known health-related quality of life (HRQoL) impact. However, no utility-based HRQoL assessments for CL patients are available. The aim of this study was to quantitatively assess the health-related quality of life among patients with CL attending a Brazilian reference center. A retrospective interview-based longitudinal study was conducted using the EQ-5D-3L/VAS to assess the current health status during active disease, and retrospectively before the onset of disease symptoms. In addition, socioeconomic data were collected via a standardized questionnaire, and sociodemographic and clinical data were collected directly from medical records. A total of 143 patients with a mean age of 52 (±17) years were included, 73% of whom were men. The mean utility score before the onset of CL symptoms was 0.858. Comparison of responses related to health status before and after disease onset revealed significant losses (p < 0.05) in all dimensions of the EQ-5D, especially those related to pain, malaise (50%) and usual activities (27%). CL also affected the median Visual Analogue Scale (VAS) score and utility score, which were 87 and 70 and 0.858 and 0.716, respectively. These results confirm the substantial negative impact of CL on all assessed dimensions of life, highlighting its role in perpetuating the cycle of suffering and poverty associated with neglected tropical diseases.

## Introduction

Cutaneous leishmaniasis (CL) is a neglected infectious parasitic disease that is widely distributed worldwide [1]. In the Americas, 37,890 CL and mucosal

**Data availability statement:** A/PROS @ ACCCEPT Please follow up for data available upon acceptance: The dataset will be published after approval of the manuscript. Previous information is available in the Mendeley repository (not published). Silva, Sarah (2025), "Health-related quality of life in Brazilian patients with cutaneous leishmaniasis using EQ-5D_dataset", Mendeley Data, V1, doi: 10.17632/mh7zrrcxcy.1"

**Funding:** Conselho Nacional de Desenvolvimento Científico e Tecnológico - CNPq (GC grant: 302069/2022-4)

**Competing interests:** The authors have declared that no competing interests exist.

leishmaniasis (ML) cases were reported in 2022, with Brazil accounting for 34% of these cases [2]. The lesions caused by CL on the skin or mucosa can result in disfiguring and stigmatizing scars. In more severe instances, these lesions may lead to functional impairments in the respiratory and digestive tracts, significantly affecting the quality of life of those impacted [3,4].

The concept of quality of life (QoL) encompasses both subjectivity and multidimensionality, incorporating various aspects, such as physical, psychological, and social dimensions [5]. In terms of an individual's perception of their emotional state, functionality, and social interactions, the assessment of QoL has become a significant health outcome to be considered alongside traditional metrics of efficacy, safety, and mortality (duration of life). The estimation of how diseases and health interventions affect QoL is understood as health-related quality of life (HRQoL) [6], a term widely used in evaluating and comparing chronic and infectious diseases such as diabetes and HIV infection [7,8]. In these contexts, beyond the duration of life, the impact on psychosocial dimensions and the overall functionality of the patient are the most relevant aspects to measure [7,9].

Studies conducted in endemic regions have explored the impact of CL on patients' QoL, using various assessment tools [3,10–22]. Instruments developed for skin diseases, such as the Dermatology Life Quality Index (DLQI) [16,17] and the Skindex-29 [10,22], are commonly used, as are tools that address psychiatric aspects [20] or even instruments specific to CL, such as the Cutaneous Leishmaniasis Impact Questionnaire (CLIQ) [11,12,15]. In addition, generic instruments such the World Health Organization Quality of Life Brief Version (WHOQOL-bref) [3] and the European Quality of Life 5 Dimensions 3 Levels (EQ-5D-3L) have been used to assess HRQoL in these patients [22].

One of the main advantages of the standardized use of generic instruments is the ability to measure health outcomes, which allows comparisons between different interventions and technologies. While disease-specific instruments, such as the DLQI, provide detailed insights into dermatological and psychosocial aspects, they do not generate utility scores necessary for economic evaluation. This, in turn, enhances the decision-making process for their incorporation into health systems. The EQ-5D-3L specifically measures HRQoL via a utility indicator that reflects social preferences for health states. One of the key outcomes in economic evaluations is Quality-Adjusted Life Years (QALYs), which consider both the dimensions of QoL and survival time in a single measure. In this regard, the aim of this study was to measure for the first time the HRQoL of patients affected by CL attending a Brazilian reference center, using the EQ-5D-3L. This instrument, developed by the EuroQol [23] group, is currently the most widely used in the world, mainly due to its simplicity and comprehensiveness, and has already been validated in Brazil [24,25].

## Methods

### Study design and data collection

This retrospective, longitudinal, interview-based study sought to include patients with CL diagnosis evaluated at a Reference Center for Leishmaniasis in Minas Gerais (MG),

Brazil, using a convenience sample of cases consecutively enrolled between October 1st, 2020 and April 30th, 2023, without predefined exclusion criteria. No formal sample size calculation was conducted prior to data collection. Instead, the sample size was determined based on feasibility and the availability of patients during the study period, with the objective of including as many eligible participants as possible. The selection criteria were an age of 12 years or older and a diagnosis of CL. Patients were consecutively recruited based on an opportunity window for an individual interview before the start of specific treatment using the EQ-5D-3L instrument (Portuguese version for Brazil, v1.0, with adaptations for each moment of the interview). Two researchers and two trained students conducted the interviews after obtaining written consent from the participants. Written informed consent was obtained from the parents or legal guardian of participants under 18 years old using Informed Consent Form (ICF), and Assent Form (AF) was obtained from the children prior to participation. During the initial contact with the patient, which occurred shortly after diagnosis (active CL), participants reported their current health status (S1 Appendix) and were asked to reflect on their condition over the previous 30 days. Given the difficulties of interviewing the entire population at risk, and in order to assess the health status of the same patients before CL onset, they were asked to retrospectively report their health condition just prior to the appearance of disease symptoms (S2 Appendix), immediately after the first interview, without specifying a timeframe. At the same time, the participants also completed a questionnaire on socioeconomic conditions (S3 Appendix). Sociodemographic and clinical information were collected directly from medical records.

## Data collection instruments

The EQ-5D-3L was used after approval from the EuroQol Group (ID 33742). The first part of the questionnaire assesses health status across five dimensions (mobility, self-care, usual activities, pain/discomfort, and anxiety/depression), each with three levels of intensity (no problems, moderate problems, and extreme problems). The combination of the five dimensions and three levels of intensity results in 243 distinct health states. These range from having no problems in any dimension (the best health state), represented by 11111, to experiencing extreme problems across all dimensions, represented by 33333. Each health state can be assigned a utility measure that reflects societal preferences for these conditions. Additionally, in the second part of the interview, participants assessed their health status via a visual analog scale (VAS), where values ranged from the worst imaginable (0) to the best (100) health state [25,26].

The socioeconomic status of the respondents was categorized according to the Brazil Economic Classification Criteria established by the Brazilian Association of Research Companies (*Associação Brasileira de Empresas de Pesquisa*— ABEP), 2019, on the basis of average family income, as follows: A (R$ 25,554.33), B1 (R$ 11,279.14), B2 (R$ 5,641.64), C1 (R$ 3,085.48), C2 (R$ 1,748.59), and DE (R$ 719.81) [27].

Sociodemographic information (sex, age, education level, municipality of residence) and clinical data (clinical form of leishmaniasis, location of the lesion, type of case—new or recurrence, presence of comorbidities, alcohol use, drug use, previous treatments) were extracted from medical records via a structured form. The age groups were categorized based on epidemiological relevance and sample distribution to ensure a balanced representation of participants across different life stages, allowing for meaningful comparisons within the study population. This study was approved by the René Rachou Institute/Fiocruz Minas's ethics committee under number CAAE: 28929220.0.0000.5091 (approval date: March 16, 2020).

## Data analysis

Each patient's health status was assessed via the EQ-5D-3L instrument, which was transformed into a utility metric. Utility values representing social preferences for health states in the Brazilian population, as estimated by Santos et al. (2016) [24], were considered (available at <https://atsinc.com.br/eq-5d-menu/>) [28]. Differences in average utility between the various subgroups, based on the patients' clinical and sociodemographic characteristics, both before CL and with active disease, were evaluated via the nonparametric Wilcoxon test. Additionally, differences in utility scores across multiple categories of clinical and sociodemographic variables were assessed using the Mann–Whitney U test (for two groups) or the Kruskal–Wallis test (for three or more groups), depending on the number of categories within each variable.

The losses in HRQoL were estimated by comparing the proportion of responses before CL onset with those during its active phase. This comparison was made across three levels of problem intensity—no problems, some problems, and extreme problems—in each of the five assessed dimensions. Differences between groups were analyzed via the Stuart–Maxwell test for marginal homogeneity of categorical variables and the Wilcoxon test for continuous variables. A significant level of 5% was established for all tests; all analyses were performed in the Statistical Package for the Social Sciences - SPSS software (v. 23).

## Results

A total of 152 patients diagnosed with CL were invited to participate, representing 47% (152/321) of the total eligible population during the study period. Of these, 143 individuals (93% of those invited) agreed to participate, with an average age of 52 years (±17) among the included patients. **Table 1** presents the general characteristics of the sample.
Most patients had the cutaneous form of leishmaniasis, including ten patients with the disseminated cutaneous form (more than six skin lesions). A total of 17 recurrences were identified, mostly in the cutaneous form, while the remaining 126 patients corresponded to new cases (new lesions, even if others had been previously treated). Among the 17 patients who experienced recurrence, six had previously undergone treatment, indicating failure of the initial therapy. In most cases, the lesions were located on the limbs. Regarding clinical characteristics, 86 patients had some concomitant disease (such as hypertension or diabetes) at the time of CL diagnosis, 20 of whom were not receiving treatment for comorbidities. Overall, the utility values were similar across the assessed categories, considering the clinical and sociodemographic characteristics of the participants. Before the onset of CL symptoms, a significant difference ($p < 0.05$) in mean utility was observed for education level. The only significant difference for average utility observed in patients with active disease was for sex ($p < 0.001$).

Before the onset of CL symptoms, the average utility was 0.858, showing a significant reduction to 0.716 after the symptoms manifested (a loss of 16.6%, $p < 0,0001$) (**Table 2**). The VAS results support the magnitude of this loss, with the average score reported by patients before CL being 87 and decreasing to 70 during active CL, representing a loss of 19.5% ($p < 0.001$).

When each dimension of the EQ-5D was analyzed before CL onset, more than half of the participants reported "no problems" across all dimensions (**Fig 1**). Following the manifestation of CL, all EQ-5D dimensions were impacted, particularly those related to pain and discomfort, as well as usual activities, where the proportion of patients reporting no issues decreased by 50% and 27%, respectively (**Fig 1** and **Table 2**).
**Table 3** presents the distributions of health states before CL onset and during the active disease phase. Prior to illness, the most common health states were those without issues in most dimensions (11111, 11121, and 11112, representing 46%, 20%, and 13%, respectively). After the onset of illness, the proportion of responses indicating no problems across all dimensions (11111) decreased to second place (16%), whereas state 11121 rose to first place (20%), and state 11122 came in third place (14%). In this last state, patients reported some issues in the dimensions of pain/discomfort and anxiety/depression, indicating a decline in the perceived health of this population.
The cumulative distribution of ordered utilities, ranging from the best to the worst health state, according to their respective utility values before and after active CL, is illustrated in **Fig 2**. The greater proportion of participants with better health states before illness is evident, as the curve for the period before CL is positioned above the curve for active CL. The difference between the curves highlights the loss of QoL associated with active CL, with the curve representing active CL reaching 100% more slowly, reflecting the presence of patients with poorer health states.

## Discussion

QALYs have been increasingly valued as a health outcome parameter in public policy decision-making because of their comprehensiveness and legitimacy in reflecting patients' perspectives on their well-being. It adds valuable information

**Table 1. Sociodemographic, economic, and clinical characteristics of patients with active cutaneous leishmaniasis treated at a Brazilian referral center between 2020 and 2023 and their impact on health-related quality of life (N=143).**

| Characteristics | N | % | Mean utility (Before CL) | *p-value* | Mean utility (Active CL) | *p-value* |
|---|---|---|---|---|---|---|
| Sex | | | | | | |
| Male | 105 | 73% | 0.877 | 0.084 | 0.749 | 0.001* |
| Female | 38 | 27% | 0.805 | | 0.626 | |
| Age | | | | | | |
| Under 25 | 12 | 8% | 0.852 | 0.492 | 0.718 | 0.925 |
| 25-49 | 41 | 29% | 0.886 | | 0.728 | |
| 50-74 | 77 | 54% | 0.849 | | 0.720 | |
| Over 74 | 13 | 9% | 0.824 | | 0.653 | |
| Region | | | | | | |
| Minas Gerais Countryside | 59 | 41% | 0.837 | 0.243 | 0.725 | 0.441 |
| Belo Horizonte | 36 | 25% | 0.868 | | 0.726 | |
| BH Border towns | 21 | 15% | 0.862 | | 0.736 | |
| Other municipalities which make up the Metropolitan Region of BH | 27 | 19% | 0.886 | | 0.666 | |
| Education level | | | | | | |
| Illiterate | 8 | 6% | 0.742 | 0.016* | 0.642 | 0.800 |
| Middle School | 60 | 42% | 0.854 | | 0.713 | |
| High School | 41 | 29% | 0.875 | | 0.717 | |
| Higher education | 24 | 17% | 0.908 | | 0.732 | |
| Not reported | 10 | 7% | 0.784 | | 0.754 | |
| Economic class | | | | | | |
| AB | 28 | 20% | 0.911 | 0.070 | 0.725 | 0.943 |
| C | 58 | 41% | 0.852 | | 0.708 | |
| DE | 57 | 40% | 0.838 | | 0.720 | |
| Comorbidity? | | | | | | |
| No | 56 | 39% | 0.887 | 0.063 | 0.742 | 0.440 |
| Yes | 86 | 60% | 0.842 | | 0.700 | |
| Not reported | 1 | 1% | 0.572 | | 0.636 | |
| Alcoholic? | | | | | | |
| No | 60 | 42% | 0.830 | 0.293 | 0.688 | 0.310 |
| Yes | 74 | 52% | 0.883 | | 0.741 | |
| Not reported | 9 | 6% | 0.840 | | 0.696 | |
| Drug user? | | | | | | |
| No | 124 | 87% | 0.853 | 0.254 | 0.713 | 0.854 |
| Yes | 10 | 7% | 0.933 | | 0.754 | |
| Not reported | 9 | 6% | 0.839 | | 0.710 | |
| New case? | | | | | | |
| No | 17 | 12% | 0.848 | 0.530 | 0.674 | 0.271 |
| Yes | 126 | 88% | 0.859 | | 0.722 | |
| Previous treatment | | | | | | |
| No | 121 | 85% | 0.855 | 0.742 | 0.721 | 0.351 |
| Yes | 22 | 15% | 0.873 | | 0.689 | |
| Clinical form | | | | | | |
| Cutaneous leishmaniasis | 116 | 81% | 0.865 | 0.270 | 0.718 | 0.353 |
| Mucous leishmaniasis | 16 | 11% | 0.805 | | 0.686 | |

*(Continued)*

**Table 1.** (Continued)

| Characteristics | N | % | Mean utility (Before CL) | p-value | Mean utility (Active CL) | p-value |
|---|---|---|---|---|---|---|
| Cutaneous and mucosal leishmaniasis | 11 | 8% | 0.854 | | 0.742 | |
| Lesion area | | | | | | |
| Lower limbs | 66 | 46% | 0.882 | 0.866 | 0.709 | 0.547 |
| Upper limbs | 32 | 22% | 0.835 | | 0.718 | |
| Head | 21 | 15% | 0.853 | | 0.746 | |
| Mucous | 16 | 11% | 0.805 | | 0.686 | |
| Trunk | 8 | 6% | 0.861 | | 0.748 | |

Legend: N: number of participants. CL: cutaneous leishmaniasis.

to the benefit of "years of life gained" from a given intervention. In Brazil, this emphasis is articulated in the Ministry of Health's Economic Evaluation Guidelines, which establish cost–utility assessments as preferred methods to support decision-making regarding the incorporation of health technologies [29]. Previous studies have shown that CL impacts the QoL of affected individuals; however, a score for use in economic analysis has not yet been developed [3,10,11,14–21,30]. In the present study, utility values for CL were achieved, enabling the future adoption of the QALYs benchmark in economic analyses of interventions for the disease.

These results indicate that the average health utility of CL patients is significantly affected by the disease. Before the onset of symptoms, participants reported an average utility of 0.858, which decreased to 0.716 during active CL. This decline aligns with findings from other studies of dermatological diseases, such as leishmaniasis, which have reported a reduction in QoL associated with various dermatologic health conditions. For instance, a European multicenter study showed lower mean VAS scores (69.9) than controls (82.2), in patients with various skin diseases [31]. Another study conducted in Vietnam assessed the impact of different skin conditions on HRQoL and found an overall mean utility score of 0.73, with the largest subgroup consisting of patients with atopic or contact dermatitis [32]. In Brazil, research on herpes zoster, a dermatological condition characterized associated with significant pain, demonstrated a decline in QoL, with utility scores dropping to 0.7 after symptoms onset [33]. These findings indicate that the decrease in HRQoL observed in CL is consistent with that seen in other skin diseases.

The present study revealed that CL impacts all the assessed domains of the EQ-5D, identifying "pain and discomfort" as the most affected dimension, characterized by the highest number of reports of "some problems" or "extreme issues" and the greatest difference compared to the period before symptom onset. Although CL typically presents as painless ulcers [34–36], other studies on CL support this finding, indicating a greater impact of CL on similar dimensions of the DLQI, such as "symptoms and feelings" [16,17,30] (understood as discomfort or embarrassment caused by the wound) [37] and specific complaints of pain [3]. This dimension, along with self-care (in the European study) and anxiety/depression (in Vietnamese patients), was also one of the most affected in studies addressing other skin diseases in Europe and Vietnam [31,32], reinforcing the importance of incorporating pain management, psychological support, and patient-centered care into clinical management.

Although this study did not find a significant difference in utilities across different lesion locations, lesions that visibly affect appearance may cause some stigma, whereas lesions on limbs may interfere with usual activities. Studies such as that conducted by Hu et al. (2020) [22] in Suriname have shown that lesions located on the lower limbs have a greater impact on HRQoL, particularly in the dimensions of self-care, mobility, usual activities, and pain and discomfort, than lesions on other parts of the body do. Facial lesions have also been associated with lower HRQoL [16] and greater psychosocial impact [21].

**Table 2. Proportion of individuals in each dimension and level of impact of the EQ-5D-3L scale before symptoms and with active cutaneous leishmaniasis.**

| EQ-5D dimensions | Moment | Intensity levels | | | | | | | *p-value* |
|---|---|---|---|---|---|---|---|---|---|
| | | No problems | | | Some problems | | Extreme problems | | |
| | | n | % | Difference* % | n (%) | % | n (%) | % | |
| **Mobility** | Before CL | 133 | 93.0 | 17.3 | 9 | 6.3 | 1 | 0.7 | <0.0001 |
| | Active CL | 110 | 76.9 | | 32 | 22.4 | 1 | 0.7 | |
| **Self-care** | Before CL | 142 | 99.3 | 15.5 | 0 | 0.0 | 1 | 0.7 | <0.0001 |
| | Active CL | 120 | 83.9 | | 22 | 15.4 | 1 | 0.7 | |
| **Habitual activities** | Before CL | 138 | 96.5 | 26.8 | 5 | 3.5 | 0 | 0.0 | <0.0001 |
| | Active CL | 101 | 70.6 | | 38 | 26.6 | 4 | 2.8 | |
| **Pain/malaise** | Before CL | 86 | 60.1 | 50 | 51 | 35.7 | 6 | 4.2 | <0.0001 |
| | Active CL | 43 | 30.1 | | 91 | 63.6 | 9 | 6.3 | |
| **Anxiety/depression** | Before CL | 101 | 70.6 | 22.8 | 34 | 23.8 | 8 | 5.6 | 0.01 |
| | Active CL | 78 | 54.5 | | 46 | 32.2 | 19 | 13.3 | |
| **Scale** | | Average score (±SD) | | – | | | | | |
| **EQ-VAS** | Before CL | 87 | (14.7) | 19.5 | – | | – | | <0.0001 |
| | Active CL | 70 | (21.6) | | | | | | |
| **EQ-5D Utility** | Before CL | 0.858 | | 16.6 | – | | – | | <0.0001 |
| | Active CL | 0.716 | | | | | | | |

Legend: CL: cutaneous leishmaniasis; EQ-VAS: Visual Analogue Scale; SD: Standard Deviation.

* Percentage difference which takes "Before CL" as 100%.

Another dimension significantly affected by CL in this study, as corroborated by other authors [3,16,17], is the ability to perform usual activities, such as working, studying, taking care of the household, meeting friends and neighbors, and attending churches. The impact on this dimension can be reflected in both the income of patients and their families, who are often already facing socio-economic disadvantages.

[13], as well as in psychosocial aspects (mental health issues and psychosocial morbidity). Some authors emphasize the importance of improving patient communication to clarify misunderstandings about the disease, which can lead to psychosocial impacts and unnecessary absences from work or school [14,18].

Anxiety and depression are also evident in the current study population, as demonstrated in previous studies [14,18–21]. These conditions represented the second dimension with the highest reports of any problems or extreme problems among patients with active CL. Notably, depression is a factor affecting HRQoL in patients with chronic diseases [38–40], and more than half of the participants in the present study had some comorbidities.

In this study, a significant difference was observed in the self-reported utility of participants, prior to disease onset, with respect to educational level, since higher educational attainment is generally associated with better economic status access to healthcare, greater financial stability, and improved overall well-being. This difference aligns with expectations and is consistent with previous studies, which has shown a correlation between higher educational attainment, employment rate, family income, and improved quality of life. [41,42]. Among patients with an active skin disease, sex was the only variable that showed a significant difference, indicating that women reported a lower perception of quality of life compared to men. This finding is consistent with studies on patients with other skin conditions. A possible explanation is that physical appearance tends to be a more sensitive concern for women than for men, and skin diseases may therefore lead to greater psychological distress among women [32,43–45].

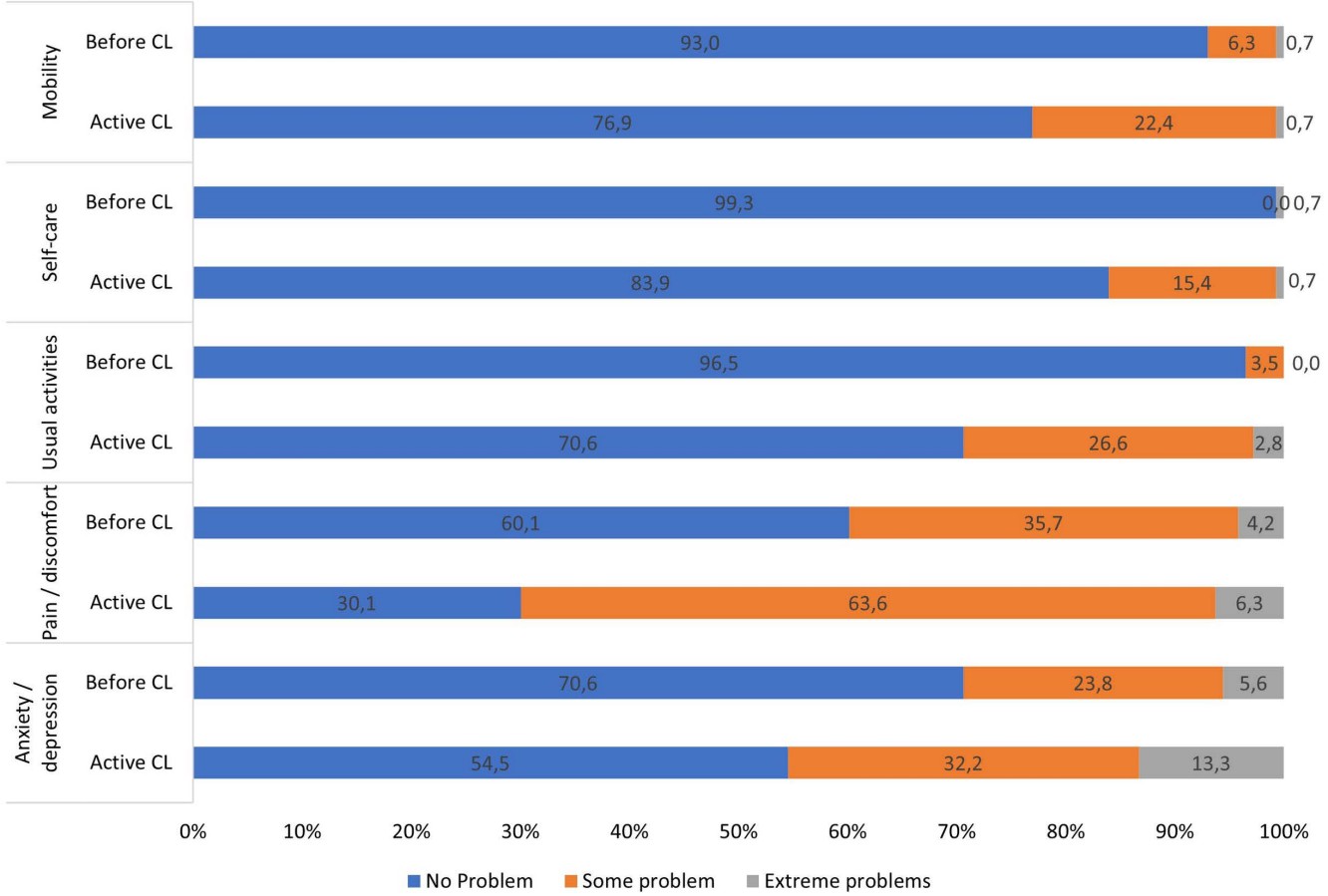

**Fig 1. Proportion of problems intensity in each dimension according to the EQ-5D-3L scale before symptoms and with active cutaneous leishmaniasis.** Legend: CL: cutaneous leishmaniasis.

This study has some limitations. The first relates to the use of generic instruments. The EQ-5D, although widely validated and useful for comparisons across different health conditions, may not capture all the nuances and particularities of CL. Despite this limitation, the instrument proved sensitive in measuring the loss of HRQoL with the onset of CL symptoms. This sensitivity was corroborated by evaluating patients via the VAS, as the magnitude of loss was similar. A second limitation pertains to the use of retrospective measures. Since patients were only monitored when they sought healthcare services due to symptoms of the disease, health states prior to the disease were lacking. Therefore, it was necessary to use retrospective measures to address this limitation, which may be subject to memory issues and changes in individuals' perceptions of their past health due to the disease's impact on their health [46,47]. Various studies [3,10,11,14–21,30], including those using the EQ-5D, have employed retrospective health measures to evaluate losses resulting from health impacts.

Additionally, a third limitation is that the inclusion of patients across a wide age range, from adolescents to elderly individuals, may introduce variability in responses, particularly in dimensions such as mobility and self-care. However, no significant differences in utility values were observed between age categories, suggesting a consistent impact of CL on HRQoL across different age groups. Nonetheless, given that baseline expectations for these domains differ significantly by age, comparisons of HRQoL loss between younger and older participants should be interpreted with caution.

**Table 3. Distribution of EQ-5D-3L health states of patients before symptoms and with active cutaneous leishmaniasis.**

| Health states | Before CL | | Active CL | |
|---|---|---|---|---|
| | N | % | N | % |
| 11111 | 66 | 0,462 | 23 | 0,161 |
| 11112 | 18 | 0,126 | 9 | 0,063 |
| 11121 | 29 | 0,203 | 29 | 0,203 |
| 11122 | 9 | 0,063 | 20 | 0,140 |
| 11123 | 3 | 0,021 | 1 | 0,007 |
| 11131 | 2 | 0,014 | 1 | 0,007 |
| 11132 | 1 | 0,007 | 0 | 0,000 |
| 11133 | 1 | 0,007 | 0 | 0,000 |
| 11211 | 1 | 0,007 | 1 | 0,007 |
| 11212 | 0 | 0,000 | 2 | 0,014 |
| 11221 | 1 | 0,007 | 1 | 0,007 |
| 11222 | 1 | 0,007 | 4 | 0,028 |
| 11223 | 0 | 0,000 | 2 | 0,014 |
| 11232 | 1 | 0,007 | 0 | 0,000 |
| 11233 | 0 | 0,000 | 4 | 0,028 |
| 12121 | 0 | 0,000 | 2 | 0,014 |
| 12122 | 0 | 0,000 | 1 | 0,007 |
| 12123 | 0 | 0,000 | 1 | 0,007 |
| 12211 | 0 | 0,000 | 1 | 0,007 |
| 12213 | 0 | 0,000 | 1 | 0,007 |
| 12221 | 0 | 0,000 | 5 | 0,035 |
| 12222 | 0 | 0,000 | 1 | 0,007 |
| 12223 | 0 | 0,000 | 1 | 0,007 |
| 21111 | 1 | 0,007 | 0 | 0,000 |
| 21112 | 0 | 0,000 | 2 | 0,014 |
| 21113 | 0 | 0,000 | 1 | 0,007 |
| 21121 | 1 | 0,007 | 5 | 0,035 |
| 21122 | 3 | 0,021 | 1 | 0,007 |
| 21123 | 2 | 0,014 | 3 | 0,021 |
| 21133 | 1 | 0,007 | 1 | 0,007 |
| 21211 | 0 | 0,000 | 2 | 0,014 |
| 21221 | 0 | 0,000 | 3 | 0,021 |
| 21222 | 1 | 0,007 | 2 | 0,014 |
| 21232 | 0 | 0,000 | 1 | 0,007 |
| 21322 | 0 | 0,000 | 1 | 0,007 |
| 21332 | 0 | 0,000 | 1 | 0,007 |
| 22121 | 0 | 0,000 | 1 | 0,007 |
| 22221 | 0 | 0,000 | 3 | 0,021 |
| 22222 | 0 | 0,000 | 1 | 0,007 |
| 22223 | 0 | 0,000 | 3 | 0,021 |
| 23311 | 0 | 0,000 | 1 | 0,007 |
| 32333 | 0 | 0,000 | 1 | 0,007 |
| 33123 | 1 | 0,007 | 0 | 0,000 |

Legend: N = number of participants; CL= cutaneous leishmaniasis

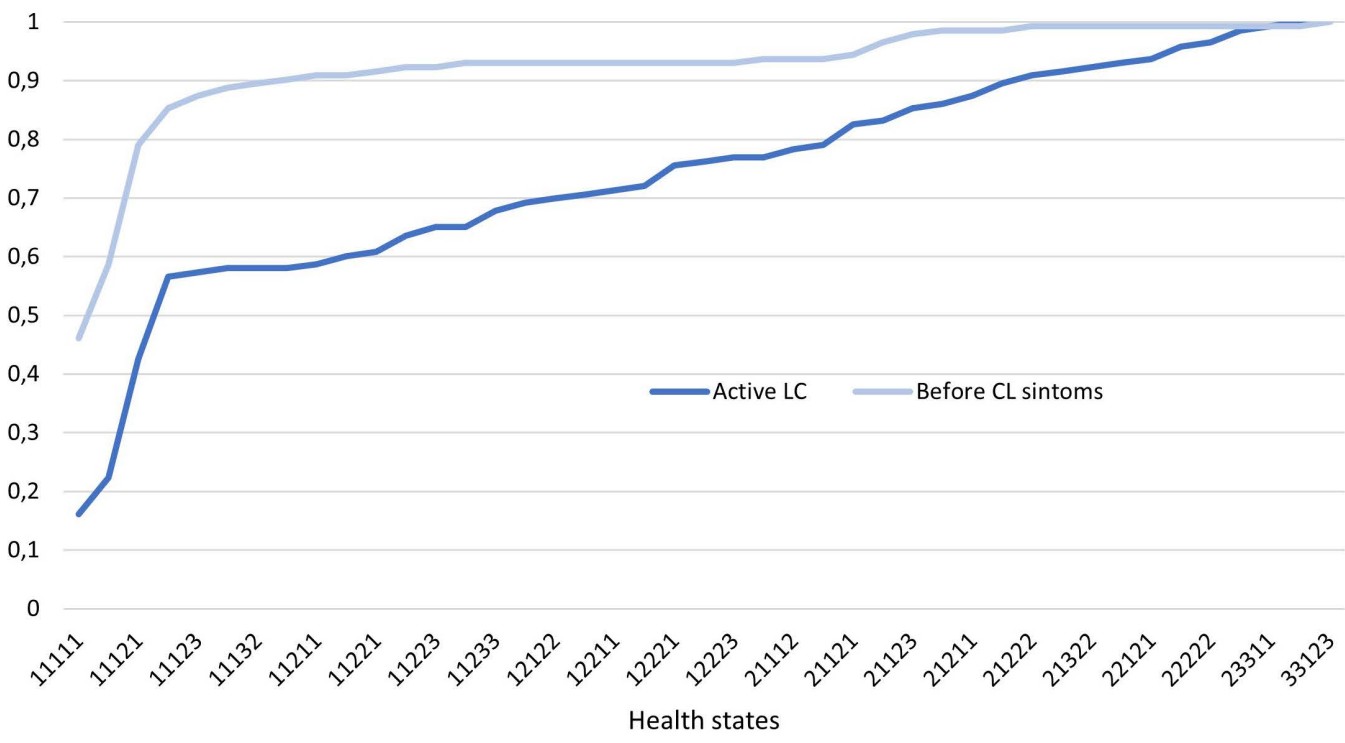

**Fig 2. Cumulative distribution of EQ-5D-3L utilities decreasingly ordered before and after the onset of cutaneous leishmaniasis symptoms.**
Legend: CL: cutaneous leishmaniasis.

The fourth limitation is related to the study design, which involved individuals from the health center conducting the interviews. Conducting the interviews at the same health center where patients receive care may introduce bias in the responses, often referred to as "gratitude bias" [48–50]. Patients may report exaggerated improvements in their QoL owing to the perception that this will please the interviewer or because they wish to be seen as cooperative.

Finally, this study may not fully represent the broader population affected by CL because it was conducted in a CL referral center, which may introduce a selection bias of more severe cases. In addition, it was based on a convenience sample rather than being derived from a calculated sample size. Patients were included based on their access to healthcare services and the availability of the researchers involved. The sample could be considered small and may lack sufficient diversity in terms of age, gender, socioeconomic status, and geographical location.

In the context of neglected diseases, the assessment of HRQoL is especially relevant, as it disproportionately affects poor and marginalized populations, increasing their susceptibility to worsening living conditions. QoL assessments can highlight the hidden burden of these diseases and justify the allocation of more resources for their control [51,52]. However, the same limitations discussed earlier apply, and the need for culturally sensitive and specific assessment tools is even more pressing.

The use of the EQ-5D in this study provided valuable insights into the impact of CL on patients' HRQoL. By generating utility measures, these findings contribute to future cost-utility analyses, allowing for evidence-based comparisons of health technologies for CL management. Moreover, this study reinforces the need for public health strategies that go beyond clinical treatment, incorporating interventions aimed at mitigating the psychosocial and economic burden of CL, particularly in vulnerable populations. From a policy perspective, these results highlight the importance of integrating HRQoL assessments into health decision-making processes, ensuring that resource allocation and intervention

prioritization consider not only disease outcomes but also patient well-being. To enhance the impact of such evaluations, future research should focus on refining assessment methodologies including the development of disease-specific utility-based instruments for more precise CL impact measurement, the improvement of data collection strategy to minimize recall bias, and the enhancement of generic instruments to better capture stigma and mental health effects. Furthermore, it is essential to prioritize expanding sample representativeness and addressing identified methodological limitations to strengthen the applicability of HRQoL data in public health decision-making and policy planning.

## Supporting information

**S1 Appendix. EQ-5D-3L – CURRENT health status (Active CL).**
(DOCX)

**S2 Appendix. EQ-5D-3L – RETROSPECTIVE health status (before CL).**
(DOCX)

**S3 Appendix. Socioeconomic questionnaire.**
(DOCX)

## Acknowledgments

The authors would like to thank the Instituto René Rachou, Fundação Oswaldo Cruz, Coordenação de Aperfeiçoamento de Pessoal de Nível Superior (CAPES), Fundação de Amparo à Pesquisa do Estado de Minas Gerais (FAPEMIG), Conselho Nacional de Desenvolvimento Científico e Tecnológico (CNPq), Centro Federal de Educação Tecnológica de Minas Gerais and Universidade Federal dos Vales do Jequitinhonha e Mucuri.

## Author contributions

**Conceptualization:** Janaína de Pina Carvalho, Sarah Nascimento Silva, Tália Santana Machado de Assis, Endi Lanza Galvão, Mônica Viegas Andrade, Kenya Valéria Micaela de Souza Noronha, Gláucia Cota.

**Data curation:** Janaína de Pina Carvalho, Sarah Nascimento Silva, Mayra Soares Moreira.

**Formal analysis:** Janaína de Pina Carvalho, Sarah Nascimento Silva, Tália Santana Machado de Assis, Endi Lanza Galvão, Mayra Soares Moreira.

**Methodology:** Janaína de Pina Carvalho, Sarah Nascimento Silva, Tália Santana Machado de Assis, Endi Lanza Galvão, Mayra Soares Moreira, Mônica Viegas Andrade, Kenya Valéria Micaela de Souza Noronha, Gláucia Cota.

**Project administration:** Janaína de Pina Carvalho, Sarah Nascimento Silva.

**Supervision:** Janaína de Pina Carvalho.

**Writing – original draft:** Janaína de Pina Carvalho, Sarah Nascimento Silva, Tália Santana Machado de Assis, Endi Lanza Galvão, Gláucia Cota.

**Writing – review & editing:** Janaína de Pina Carvalho, Sarah Nascimento Silva, Tália Santana Machado de Assis, Endi Lanza Galvão, Mayra Soares Moreira, Mônica Viegas Andrade, Kenya Valéria Micaela de Souza Noronha, Gláucia Cota.

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
