## [Decision Letter · Decision Letter 0]

24 Feb 2025

PONE-D-24-57829Health-related quality of life in patients with cutaneous leishmaniasis in Brazil using EQ-5DPLOS ONE

Dear Dr. Silva,

Thank you for submitting your manuscript to PLOS ONE. After careful consideration, we feel that it has merit but does not fully meet PLOS ONE’s publication criteria as it currently stands. Therefore, we invite you to submit a revised version of the manuscript that addresses the points raised during the review process.

Please consider the points raised by the reviewers. I emphasize the need for additional statistical analyses to adequately address the concerns raised by Reviewer #1.

We look forward to receiving your revised manuscript.

Kind regards,

Vinícius Silva Belo

Academic Editor

PLOS ONE

2. Please provide additional details regarding participant consent. In the ethics statement in the Methods and online submission information, please ensure that you have specified (1) whether consent was informed and (2) what type you obtained (for instance, written or verbal, and if verbal, how it was documented and witnessed). Since your study included minors, state whether you obtained consent from parents or guardians. If the need for consent was waived by the ethics committee, please include this information.

“Conselho Nacional de Desenvolvimento Científico e Tecnológico - CNPq (GC grant:

302069/2022-4)”

4. In the online submission form, you indicated that your data is available only on request from a third party. Please note that your Data Availability Statement is currently missing the name of the third party contact or institution / contact details for the third party, such as an email address or a link to where data requests can be made. Please update your statement with the missing information.

Reviewers' comments:

Reviewer's Responses to Questions

**Comments to the Author**

1. Is the manuscript technically sound, and do the data support the conclusions?

Reviewer #1: Partly

Reviewer #2: Partly

Reviewer #3: Yes

2. Has the statistical analysis been performed appropriately and rigorously? 

Reviewer #1: No

Reviewer #2: Yes

Reviewer #1: Yes

3. Have the authors made all data underlying the findings in their manuscript fully available?

Reviewer #1: Yes

Reviewer #2: Yes

Reviewer #3: Yes

4. Is the manuscript presented in an intelligible fashion and written in standard English?

Reviewer #1: Yes

Reviewer #2: Yes

Reviewer #3: Yes

5. Review Comments to the Author

Reviewer #1: Evaluation of the article entitled: Health-related quality of life in patients with cutaneous leishmaniasis in Brazil using EQ-5D

Overall impression

The topic/problem addressed is highly relevant and needs to be shared. However, the article presents methodological weaknesses that limit its publication, including the following: the study design is described as cross-sectional; however, the methodology presents the same comparison variables within the same population at different moments (before the CL diagnosis and after the onset of the first symptoms), which would characterize it as longitudinal.

Another highly relevant aspect is the uniform use of questionnaires to assess "quality of life," which involve variables such as, mobility and self-care, among people of widely varying age groups. These variables (mobility and self-care) are categorized differently across age groups. For example, I do not think it is appropriate to compare the mobility of a 12-year-old person with that of someone over 74 years old.

Moreover, the exclusion criteria were not clearly defined. For instance, wouldn't a person with a prior diagnosis of depression constitute a confounding bias? Furthermore, the results do not clearly indicate whether the "decline" in quality of life was independently associated with the onset of CL symptoms or if it could have been influenced by other health conditions, like as long-standing diabetes mellitus.

Analysis by items

Abstract:

✔Line 27: Review the phrase: "in patients with CL in patients." There is a repetition of terms ("in patients").

✔Objective: I suggest removing "in patients attending a Brazilian reference center, using the EQ-5D-3L/VAS questionnaire to generate utility measures for this disease," as this information is better suited for the "Methods" section.

Methods:

✔ the study design is described as cross-sectional; however, the methodology presents the same comparison variables within the same population at different moments (before the LC diagnosis and after the onset of the first symptoms), which would characterize it as longitudinal.

Results:

✔The sentence "The mean utility score before the onset of CL symptoms was 0.858. A comparison of the responses related to health status before and after the illness revealed significant losses (p<0.05) in all dimensions of the EQ-5D, especially those related to pain, malaise, and usual activities, which were reduced to 50% and 27%, respectively" seems paradoxical. The way it is written suggests a reduction in pain, malaise, and activities. I suggest rewording this sentence.

Article text

✔Introduction

I suggest explicitly stating the objective in the introduction.

✔Methods:

Study design: Considering that the same individuals were evaluated with the same variables at different time points (before the onset of symptoms and during the active phase of CL), I suggest revising the study design to be described as longitudinal.

I suggest clarifying the variable used to assess health status before and during the active phase of CL. For example: How was the timeframe determined to define the period before symptom onset? (One week before? One month before? One year before?). Regarding the active disease phase, it is crucial to specify that it was 30 days after symptom onset.

The age range of participants varied from adolescence (12 years and older) to elderly individuals (over 74 years old). It would be essential to explain the rationale for selecting this age range because quality of life is sensitive to age. For example, mobility or self-care (as evaluated in the questionnaire) before symptom onset and during the active disease phase may not be comparable between a 12-year-old individual and someone over 74.

It would be essential to clarify whether an analysis was conducted to assess quality of life in association with socioeconomic, sociodemographic, and clinical variables, such as age and comorbidities.

I suggest further clarifying the inclusion criteria. For example, were individuals with a prior diagnosis of depression and/or anxiety included in the study?

✔Results:

I believe it would be more objective to concisely describe the convenience sample relative to the total eligible population.

The statement "The VAS results support the magnitude of this loss, with the average score reported by patients before CL being 87 and decreasing to 70 during active CL, representing a loss of 19.5%" needs a clearer statistical description to demonstrate that this score reduction on the VAS was significant and independently associated with the CL diagnosis. (It is stated that more than 50% of participants had comorbidities such as diabetes and hypertension, some of whom were untreated, which could be a factor affecting vision.)

I suggest avoiding repetition of information from the table in the text, as it becomes redundant. Instead, keep only complementary details in the text.

I recommend completing the table titles so that they can be interpreted and understood independently of the text. For example, missing details include location and study period.

The age groups in the results table were categorized in a non-standard manner. I suggest explaining this categorization in the Methods section or providing a reference to justify it. I reinforce that the broad age range may impact quality of life differently among individuals with CL.

The figures are not clear; I suggest improving their resolution for better readability.

Reviewer #2: Its interesting manuscript about the "Health-related quality of life in patients with cutaneous leishmaniasis in Brazil using EQ-5D".

The manuscript presents a well-structured study on the impact of cutaneous leishmaniasis (CL) on health-related quality of life (HRQoL) using the EQ-5D-3L instrument in a Brazilian cohort. The study addresses an important gap in the literature by quantitatively assessing the disease burden using standardized measures. The methodology is sound, and the statistical analyses are appropriate. However, there are some areas that require improvement in structure, clarity, and interpretation of results.

1. Title and Abstract

Title: The title is clear and informative but could be slightly refined for conciseness. Suggested revision:

"Health-Related Quality of Life in Brazilian Patients with Cutaneous Leishmaniasis Using EQ-5D-3L".

Abstract: The abstract is well-structured but contains minor grammatical errors and awkward phrasing.

The sentence "The aim of the present study was quantitatively assess the health-related quality of life..." should be rewritten as:

"The aim of this study was to quantitatively assess the health-related quality of life..."

The conclusion could be more impactful by emphasizing the broader implications for public health and policy.

2. Introduction

The introduction provides a strong rationale for the study, outlining the burden of CL and the importance of HRQoL assessment.

The references are relevant, but the discussion of previous studies could be slightly condensed to improve readability.

The sentence "However, no studies were found that quantitatively defined the health utility of CL." might be misleading because studies have assessed QoL in CL patients using different instruments. Instead, it should specify that no utility-based HRQoL assessments had been conducted.

3. Methods

The study design and statistical approach are well-described.

The retrospective assessment of pre-disease HRQoL introduces recall bias, which is acknowledged, but additional justification on why this approach was necessary could strengthen the discussion.

The use of EQ-5D-3L is appropriate, but a brief mention of why this tool was preferred over disease-specific QoL instruments (e.g., Dermatology Life Quality Index) would be beneficial.

The sample size justification is missing—was a power calculation conducted?

Recommendation: Moderate revision to clarify study design choices and add justification for the sample size.

4. Results

The results are clearly presented with appropriate statistical analyses.

Figures and tables are well-organized but should be cross-referenced more explicitly in the text.

The finding that pain/discomfort and usual activities were the most affected dimensions is expected but could be better contextualized with references to similar conditions (e.g., post-herpetic neuralgia, chronic skin conditions).

There is no discussion of potential sex-based differences despite reporting a significant difference in baseline HRQoL scores between men and women.

Recommendation: Moderate revision to improve result interpretation and discuss observed differences more thoroughly.

5. Discussion

The discussion effectively summarizes key findings and places them in context.

The authors compare their utility scores with those from studies on other diseases (e.g., HIV, herpes zoster), which is useful but should be expanded to explain why these comparisons are relevant.

Some speculative statements require more evidence, e.g., "These findings underscore the severity of the impact of CL on QoL, which is comparable to that of other chronic and debilitating diseases."—This should be supported by additional references or phrased more cautiously.

The limitations section is comprehensive, but it does not address potential selection bias—since patients were recruited from a referral center, they may represent more severe cases of CL.

Recommendation: Major revision to strengthen discussion points and refine conclusions.

6. Conclusion

The conclusion aligns with the study’s findings but could be more impactful by emphasizing the implications for clinical care and policymaking.

The sentence "To maximize the value of these studies, it is crucial to continue enhancing assessment instruments..." is vague—what specific improvements are suggested?

Recommendation: Minor revision for a stronger takeaway message.

Final Decision

☑ Major Revision Required

Justification: While the study is methodologically sound and addresses an important research gap, it requires significant refinements in the discussion, result interpretation, and methodological justification. The authors should focus on improving clarity, addressing potential biases, and strengthening their argumentation.

Once the revisions are completed, the manuscript is likely to be suitable for publication.

Reviewer #3: The manuscript provides valuable insights into the health-related quality of life (HRQoL) in cutaneous leishmaniasis (CL) patients using the EQ-5D-3L/VAS questionnaire. The study is well-designed, and the data appropriately support the conclusions. However, some minor revisions are recommended:

**Clarity & Readability:**  Some sentences need rewriting . For example, “The aim of the present study was quantitatively assess…” should be revised to “The aim of this study was to quantitatively assess…”. Consider reviewing sentence structures for clarity.**Terminology:**  Replace “transversal study” with “cross-sectional study” for better clarity, as the latter is the standard term in English-language research.

Overall, the manuscript is of high quality and makes a valuable contribution to the understanding of CL’s impact on HRQoL. With minor revisions for clarity and precision, it will be suitable for publication.  

I have no competing interest

No body assisted me in preparing this review

6. PLOS authors have the option to publish the peer review history of their article (what does this mean? ). If published, this will include your full peer review and any attached files.

**Do you want your identity to be public for this peer review?** For information about this choice, including consent withdrawal, please see our Privacy Policy .

Reviewer #1: No

Reviewer #2: **Yes: ** Mehdi Azami

Reviewer #3: Shimelis Nigusse Doni

---

## [Author Response · Author response to Decision Letter 1]

3 Apr 2025

Dear reviewers

We would like to thank the reviewers for their careful review of the manuscript "Health-related quality of life in patients with cutaneous leishmaniasis in Brazil using EQ-5D" (PONE-D-24-57829), now with the changed title “Health-related quality of life in Brazilian patients with cutaneous leishmaniasis using EQ-5D”. This opportunity to improve our work makes it even more robust and useful to the scientific community - our sole goal.

We did our best to improve the text accordingly. The modifications implemented in this revised version are summarized below together with specific answers to the questions placed by the reviewers.

We reinforced that the funders had no role in study design, data collection and analysis, decision to publish, or preparation of the manuscript.

We hope the reviewers find our responses to their comments satisfactory. Please let us know if any further improvements to the manuscript are needed.

Sincerely yours,

Sarah Silva

---

## [Decision Letter · Decision Letter 1]

2 May 2025

Health-related quality of life in Brazilian patients with cutaneous leishmaniasis using EQ-5D

PONE-D-24-57829R1

Dear Dr. Silva,

We’re pleased to inform you that your manuscript has been judged scientifically suitable for publication and will be formally accepted for publication once it meets all outstanding technical requirements.

Kind regards,

Vinícius Silva Belo

Academic Editor

PLOS ONE

Additional Editor Comments (optional):

Reviewers' comments:

Reviewer's Responses to Questions

**Comments to the Author**

1. If the authors have adequately addressed your comments raised in a previous round of review and you feel that this manuscript is now acceptable for publication, you may indicate that here to bypass the “Comments to the Author” section, enter your conflict of interest statement in the “Confidential to Editor” section, and submit your "Accept" recommendation.

Reviewer #1: All comments have been addressed

Reviewer #2: All comments have been addressed

2. Is the manuscript technically sound, and do the data support the conclusions?

Reviewer #1: Yes

Reviewer #2: Yes

3. Has the statistical analysis been performed appropriately and rigorously? 

Reviewer #1: Yes

Reviewer #2: Yes

4. Have the authors made all data underlying the findings in their manuscript fully available?

Reviewer #1: Yes

Reviewer #2: Yes

5. Is the manuscript presented in an intelligible fashion and written in standard English?

Reviewer #1: Yes

Reviewer #2: Yes

6. Review Comments to the Author

Reviewer #1: I agree with approving the article after the authors have revised it. They were careful and made the requested changes, from format to meet the plos one guidelines to weaknesses that compromised the excellence of the research, including changing the study design.

Reviewer #2: Dear authors,

The article has been revised and corrected as requested. At this stage, I consider it acceptable for publication and will make a recommendation to the editor as to its suitability.

7. PLOS authors have the option to publish the peer review history of their article (what does this mean? ). If published, this will include your full peer review and any attached files.

**Do you want your identity to be public for this peer review?** For information about this choice, including consent withdrawal, please see our Privacy Policy .

Reviewer #1: No

Reviewer #2: **Yes: ** Mehdi Azami

---

## [Editor Report · Acceptance letter]

PONE-D-24-57829R1

PLOS ONE

Dear Dr. Silva,

I'm pleased to inform you that your manuscript has been deemed suitable for publication in PLOS ONE. Congratulations! Your manuscript is now being handed over to our production team.

Kind regards,

on behalf of

Dr. Vinícius Silva Belo

Academic Editor

PLOS ONE